# Reverse transcriptase inhibitors enable the generation of fertile spermatids from fetal mouse testes in vitro
Mayuka Nishida[1], Yukina Ono-Sunagare[1], Sayuri Kato[1], Yu Ishikawa-Yamauchi [2], Takafumi Matsumura[2], Mitsuru Komeya[3], Shogo Matoba [4], Kimiko Inoue [4], Narumi Ogonuki[4], Atsuo Ogura[4], Takehiko Ogawa [2] & Takuya Sato [2] ✉

Organ culture systems enabling in vitro spermatogenesis from neonatal mouse testes exist, but differentiation from fetal testes shortly after sex determination remains unsuccessful. Here, we report the in vitro generation of fertile haploid cells from E12.5 fetal testes. While optimizing in vitro spermatogenesis protocols for neonatal testes, we find that supplementing the culture medium with reverse transcriptase inhibitors (RTIs) significantly improves the efficiency of spermatogenesis, by suppressing retrotransposon activity and protecting genomic integrity. Applying this approach, we successfully induce spermatogenesis through to the elongating spermatid by culturing E12.5 fetal testes under hypoxic conditions in RTI-supplemented medium. Notably, microinsemination using in vitro-derived spermatids produces healthy and fertile offspring, confirming their functional competence. These findings demonstrate the faithful in vitro recapitulation of testicular development and complete spermatogenesis from an early fetal stage, providing a valuable platform for investigating early germ cell development and reproductive biology.

Using the gas-liquid interface culture method, we previously succeeded in differentiating fertile haploid cells in vitro from neonatal mouse testicular tissue fragments[1]. Subsequently, this method has also been successfully used to induce complete in vitro spermatogenesis from other types of tissues, including adult mouse and cryopreserved testicular tissues[2,3].

Despite these advances, the efficiency of spermatogenesis in testicular organ culture remains markedly lower than in vivo, particularly when using fetal testis tissue. Although spermatid formation has been achieved in vitro using fetal testes, the efficiency declines significantly with earlier developmental stage. Notably, in organ cultures of testes from E12.5–13.5, spermatogenesis is rarely, if ever, induced[4]. This developmental window is characterized by dynamic morphological and molecular transitions. In mice, the fetal testis originates from the genital ridge at E10.5. By E11.5, the gonad and mesonephros become distinguishable, although still sexually undifferentiated. Sex determination occurs around E12.5, leading to Sertoli cell differentiation and testis cord formation, which permits morphological distinction between male and female gonads[5]. Thereafter, the testis undergoes further maturation, with elongation of testis cords and the appearance of Leydig and myoid cells, completing major structural formation by E14.5[6].

Concurrently, germ cells undergo profound epigenetic reprogramming. DNA methylation levels in primordial germ cells decline to a minimum around E12.5–13.5, coinciding with sex determination, and begin rising again after E13.5[7].

In this study, we investigated whether in vitro spermatogenesis could be initiated from E12.5 fetal testes, developmentally prior to the point at which successful culture has been previously achieved, and report conditions that enabled progression from testicular development to spermatid formation in vitro.

## Results

### Reverse transcriptase inhibitors (RTIs) enhance in vitro spermatogenesis efficiency in neonatal mouse testes

*Acrosin* (*Acr*)-GFP mice, which express GFP in spermatogenic cells from the mid-meiotic phase onward, serve as a valuable tool for evaluating and optimizing in vitro spermatogenesis using neonatal mouse testes. To enhance the efficiency of in vitro spermatogenesis, we screened various culture medium supplements, including growth factors and functional small molecules. During this process, we noticed that RTIs appeared to promote

[1]Laboratory of Biopharmaceutical and Regenerative Sciences, Institute of Molecular Medicine and Life Science, Yokohama City University Association of Medical Science, Yokohama, Kanagawa, Japan. [2]Department of Regenerative Medicine, Yokohama City University Graduate School of Medicine, Yokohama, Kanagawa, Japan. [3]Department of Urology, Yokohama City University School of Medicine, Yokohama, Kanagawa, Japan. [4]BioResource Research Center, RIKEN, Tsukuba, Ibaraki, Japan. ✉e-mail: tsato@yokohama-cu.ac.jp

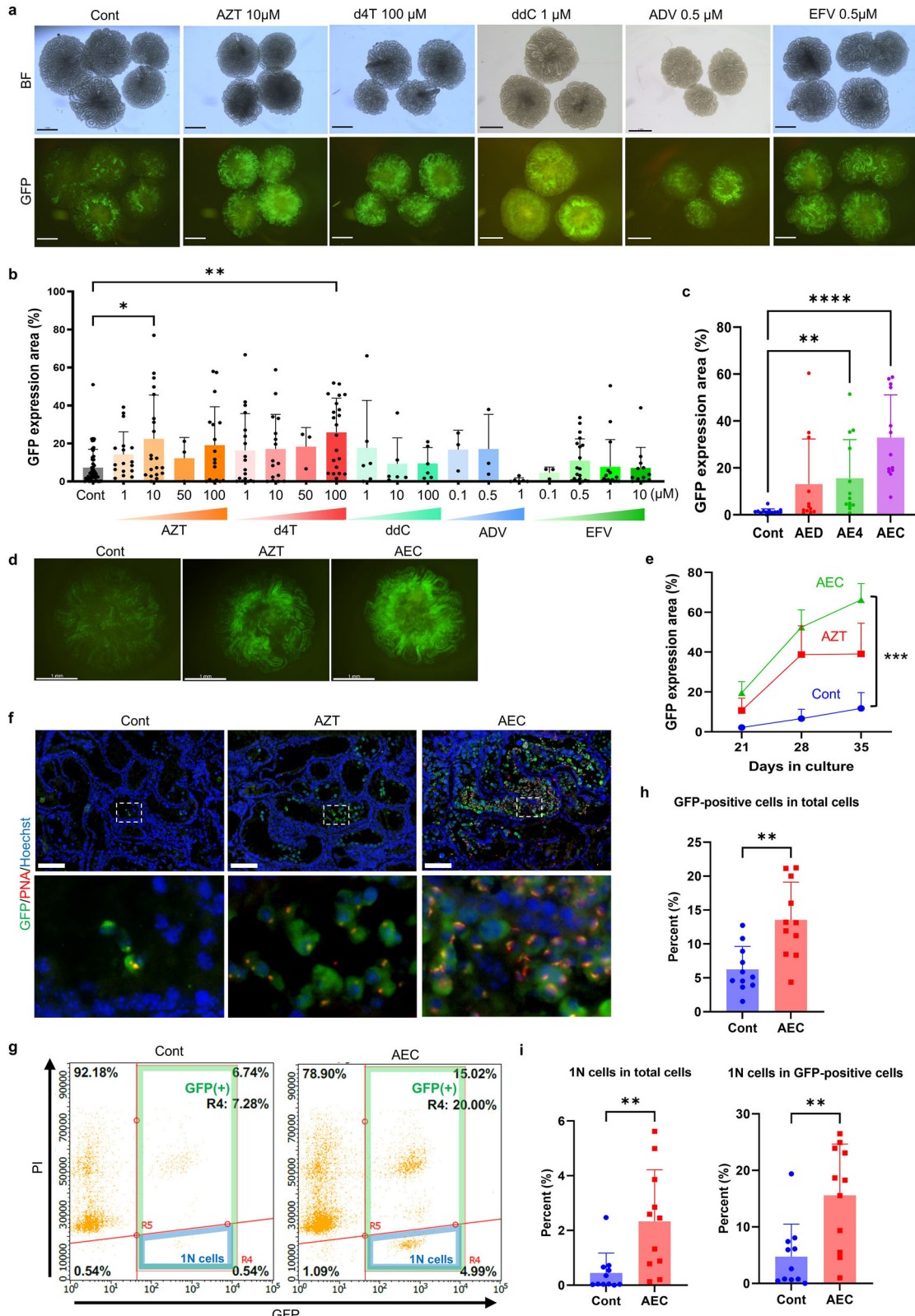

in vitro spermatogenesis when added to the culture medium, prompting further screening. We then conducted screening experiments using several RTIs. In the initial screen, we observed that azidothymidine (AZT) and stavudine (d4T), when used individually, increased GFP expression (Fig. 1a, b). More notably, a combination of three RTIs with distinct mechanisms of action further enhanced the induction of GFP-positive areas, suggesting a

synergistic effect. This combination, hereafter referred to as AEC, consisted of AZT (10 μM), efavirenz (EFV, 0.1 μM), and dideoxycytidine (ddC, 100 μM) (Fig. 1c).

To validate the effect of AEC, we conducted a time-course analysis of GFP-positive areas in AZT-alone and AEC-supplemented groups on days 21, 28, and 35 of culture (Fig. 1d, e). By day 28, both groups showed

**Fig. 1 | Effects of RTI supplementation on in vitro spermatogenesis in neonatal mouse testes. a** Representative stereomicroscope images of testicular tissues cultured in control medium (Cont) and media containing various RTIs. Testes from 1.5–3.5-day-old *Acr*-GFP mice were cultured for 36–40 days. At the end of culture, 3–5 tissue fragments per condition were transferred to a culture dish, and both brightfield (BF) and GFP fluorescence images were captured. RTIs used include: AZT (azidothymidine), d4T (stavudine), ddC (dideoxycytidine), ADV (adefovir), and EFV (efavirenz). Scale bars = 1 mm. **b** Quantification of GFP-positive area in testicular tissues cultured with each RTI. Testes from 1.5–5.5-day-old mice were cultured for 36–42 days. The GFP-positive area was measured and expressed as a percentage of the total tissue area (mean ± s.d.). *$p < 0.05$, **$p < 0.005$ (Dunn's multiple comparisons test). **c** GFP-positive area in testes cultured with combinations of RTIs. Three RTI mixes: AED (AZT + EFV + d4T), AE4 (AZT + EFV + ddC), and AEC (AZT + EFV + ddC) were compared. **$p = 0.0028$, ****$p < 0.0001$ (Dunn's multiple comparisons test). **d** Representative GFP fluorescence images of testicular tissues cultured for 35 days with control medium, AZT alone, or AEC combination. Scale bar = 1 mm. **e** Time-course analysis of GFP-positive area in tissues cultured with Cont, AZT, or AEC. $n = 5$. ***$p = 0.0005$ (Dunn's multiple comparisons test). **f** Immunofluorescence staining of tissues cultured with RTI combinations. Sections were labeled with anti-GFP antibody (green), acrosome marker peanut agglutinin (PNA; red), and counterstained with Hoechst 33342 (blue). Enlarged views of the boxed regions are shown below. Scale bar = 100 μm. **g** Representative flow cytometry plot of cultured testicular cells stained with anti-GFP antibody and propidium iodide (PI). The horizontal axis represents GFP fluorescence intensity, and the vertical axis represents PI fluorescence. Region R4 indicates GFP-positive cells; within this, region R5 (lower right quadrant) represents haploid GFP-positive cells. **h, i** Quantification of germ cells by flow cytometry. **h** Proportion of GFP-positive cells among total cells. **i** Left: percentage of haploid (1 N) cells among all cells; right: percentage of haploid cells among GFP-positive cells. **$p < 0.01$ (Mann–Whitney test).

increased GFP-positive area compared to the control. By day 35, the AEC group exhibited significantly more extensive GFP expression than the AZT-alone group, with nearly a 1.7-fold increase.

Histological analysis of cultured tissues further supported these observations. Immunostaining with an anti-GFP antibody revealed abundant GFP-positive cells in both groups, with the AEC group displaying a more robust signal. Peanut agglutinin (PNA), a marker for the acrosome of haploid cells, labeled numerous mature cells in the AEC group, indicating more advanced spermatogenic progression, including the presence of round and elongated spermatids (Fig. 1f).

To further quantify the effect of AEC, we performed flow cytometric analysis. Cultured tissues were enzymatically dissociated, fixed, and stained with anti-GFP antibody and propidium iodide (PI) (Fig. 1g). AEC treatment roughly doubled the proportion of GFP-positive cells compared to control (Fig. 1h). Moreover, the proportion of haploid cells was markedly increased; among GFP-positive cells, the haploid population more than tripled with AEC supplementation (Fig. 1i).

These findings demonstrate that RTI supplementation enhances in vitro spermatogenesis in neonatal mouse testes, with the AEC combination exhibiting especially strong efficacy in promoting both meiotic entry and post-meiotic progression.

## Effects of RTIs on in vitro spermatogenesis from E14.5 fetal testes

Our previous study demonstrated that the efficiency of in vitro spermatogenesis in fetal testes at E18.5-19.5 (just prior to birth) was comparable to that in neonatal testes. However, testes at earlier developmental stages showed significantly reduced efficiency, with spermatogenesis failing to occur in testes younger than E13.5[4]. Therefore, in the present study, we used E14.5 fetal testes to examine the effects of RTIs on in vitro spermatogenesis. As previously reported, GFP expression in the control group was limited to small areas[4]. AZT treatment moderately increased the GFP-positive area, whereas the AEC led to a much broader distribution of GFP-expressing areas (Fig. 2a, b). To further evaluate spermatogenic progression, we performed immunostaining on cryosections of cultured tissue to detect PNA-positive spermatids. While no PNA-positive spermatids were detected in the control group, a small number were observed in the AZT-treated group. In contrast, the AEC group showed a markedly greater number of PNA-positive spermatids (Fig. 2c). These results suggest that RTI supplementation improves the efficiency of in vitro spermatogenesis in fetal testes. These findings extend the applicability of RTIs to fetal testes at E14.5, with AEC demonstrating superior efficacy to AZT.

## Culture of E12.5 fetal testes using RTIs-supplemented medium

Given that spermatogenesis has never been induced from fetal testes at E12.5 in vitro[4], we next investigated whether RTI supplementation could support germ cell differentiation at this earlier developmental stage. E12.5 testes were cultured using the same gas-liquid interface system as in previous experiments, with media supplemented with the AEC combination. GFP

expression and histological analyses were used to assess spermatogenic progression.

During the culture period, testicular cords developed into seminiferous tubule-like structures in both the control and AEC-supplemented groups (Fig. 3a). Notably, localized GFP expression was observed in 5 of 9 AEC-treated testes, and immunostaining confirmed the presence of acrosome-bearing spermatids, as indicated by PNA staining (Fig. 3b). These findings demonstrate that RTI supplementation can induce spermatogenesis in vitro even from the E12.5 fetal stage. However, the induction efficiency appeared limited, as GFP expression was restricted to discrete areas. Additionally, testicular growth in the AEC group was reduced compared to control, suggesting that impaired growth may be constraining spermatogenic progression under these conditions.

We hypothesized that enhancing testicular growth during the initial culture period might improve the efficiency of spermatogenic induction. Given that AEC induced spermatogenesis efficiency from E14.5 testes, we modified the culture conditions for the first two days (referred to as "1st Medium," from E12.5 to the equivalent of E14.5) while maintaining the standard AEC medium thereafter. To promote early testicular growth, our basal medium was supplemented with Fetal Bovine Serum (FBS) instead of KnockOut serum Replacement (KSR). We tested at two concentrations: 2.5% (F2.5) and 10% (F10). Both FBS conditions showed improved growth relative to the AEC group, suggesting partial alleviation of growth inhibition (Fig. 3c). However, by week 7, analysis of the GFP-positive areas revealed no marked improvement under the FBS-containing conditions (Fig. 3d). Furthermore, testis tissues cultured with FBS contained fewer PNA-positive cells compared to those cultured with AEC alone (Fig. 3e, f and Supplementary Fig. 1). These results indicate that, although FBS-based approaches partially alleviated growth inhibition, they did not improve spermatogenic induction. Alternative strategies to support early testicular growth may further enhance efficiency, but at present the AEC condition remains the most effective for inducing spermatogenesis from E12.5 testes.

## In vitro spermatogenesis from fetal testes under hypoxic conditions

We previously demonstrated that culturing neonatal mouse and rat testes under hypoxic conditions improves in vitro spermatogenesis efficiency[8–10]. Based on these findings, we examined whether hypoxia could similarly enhance in vitro spermatogenesis from fetal testes. We compared cultures maintained at 20% $O_2$ versus 10% $O_2$ using two media conditions: control and AEC-supplemented. Both control and AEC conditions showed more extensive GFP expression and improved spermatogenesis efficiency at 10% oxygen compared to 20% oxygen (Fig. 4a, b). Notably, the AEC condition at 10% oxygen yielded the broadest GFP expression area and increased spermatogenesis efficiency to approximately 33% of the tissue. An improvement was observed not only in the proportional GFP expression area but also in the absolute expression area (Supplementary Fig. 2a).

Immunohistochemical analysis showed that even the control condition, without AEC, at 10% oxygen produced a small number of PNA-

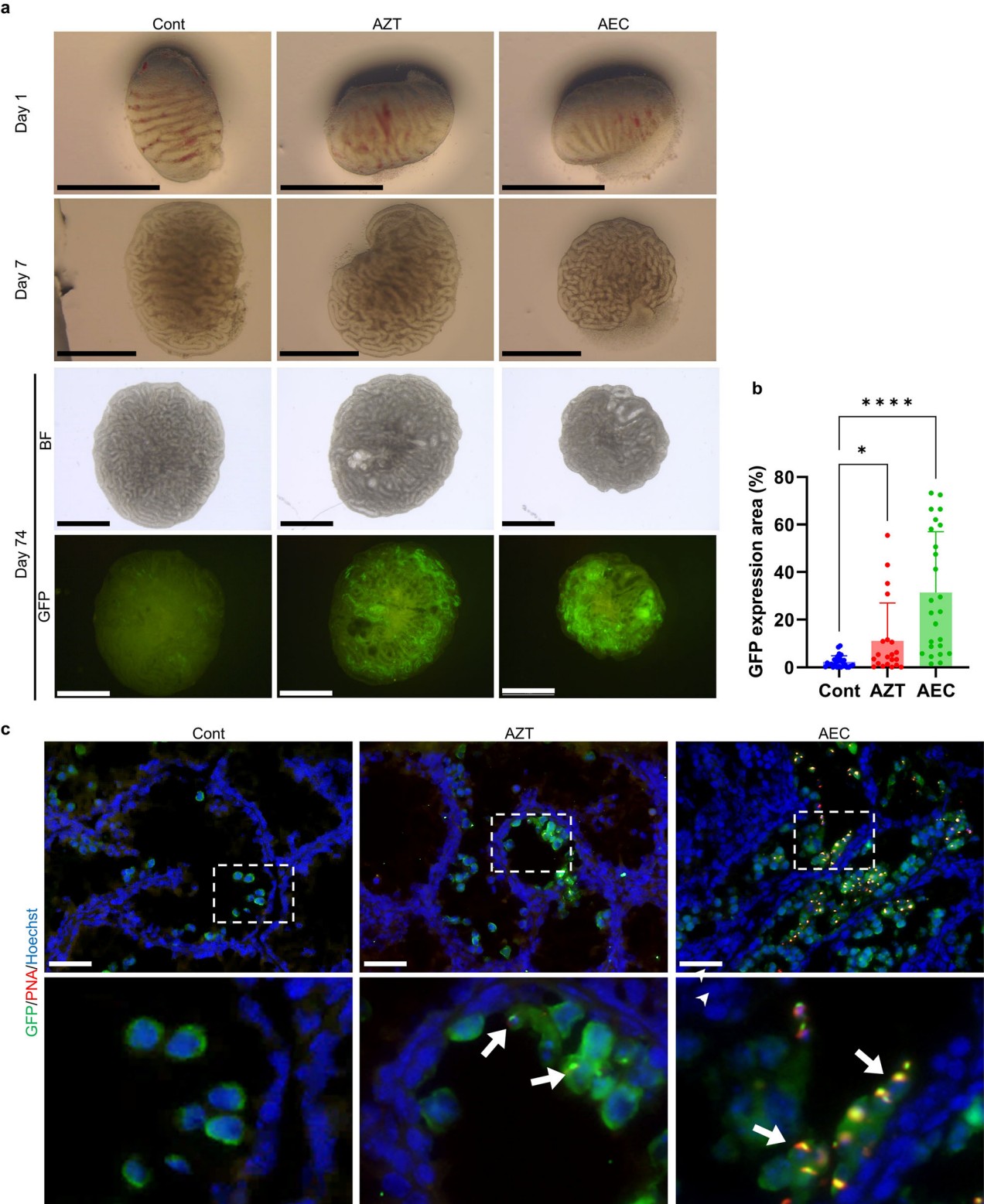

**Fig. 2 | In vitro spermatogenesis from E14.5 fetal testes culture with RTIs.**
**a** Representative streomicroscope images of fetal testicular tissues (E14.5) cultured in RTI-supplemented media, shown at days 1, 7, and 74 of culture. Scale bar = 1 mm.
**b** Quantification of GFP-positive area after 7 weeks of culture with or without RTIs. Data are presented as mean ± s.d. *$p = 0.0423$, ****$p < 0.0001$ (Dunn's multiple comparison test). **c** Immunofluorescence analysis of fetal testes cultured for 56 days. Sections were stained with anti-GFP antibody (green), PNA (red), and Hoechst 33342 (blue). Enlarged views of boxed regions in the upper panels are shown below. Arrows indicate round spermatids observed in the AZT- and AEC-treated groups. Scale bar = 50 μm.

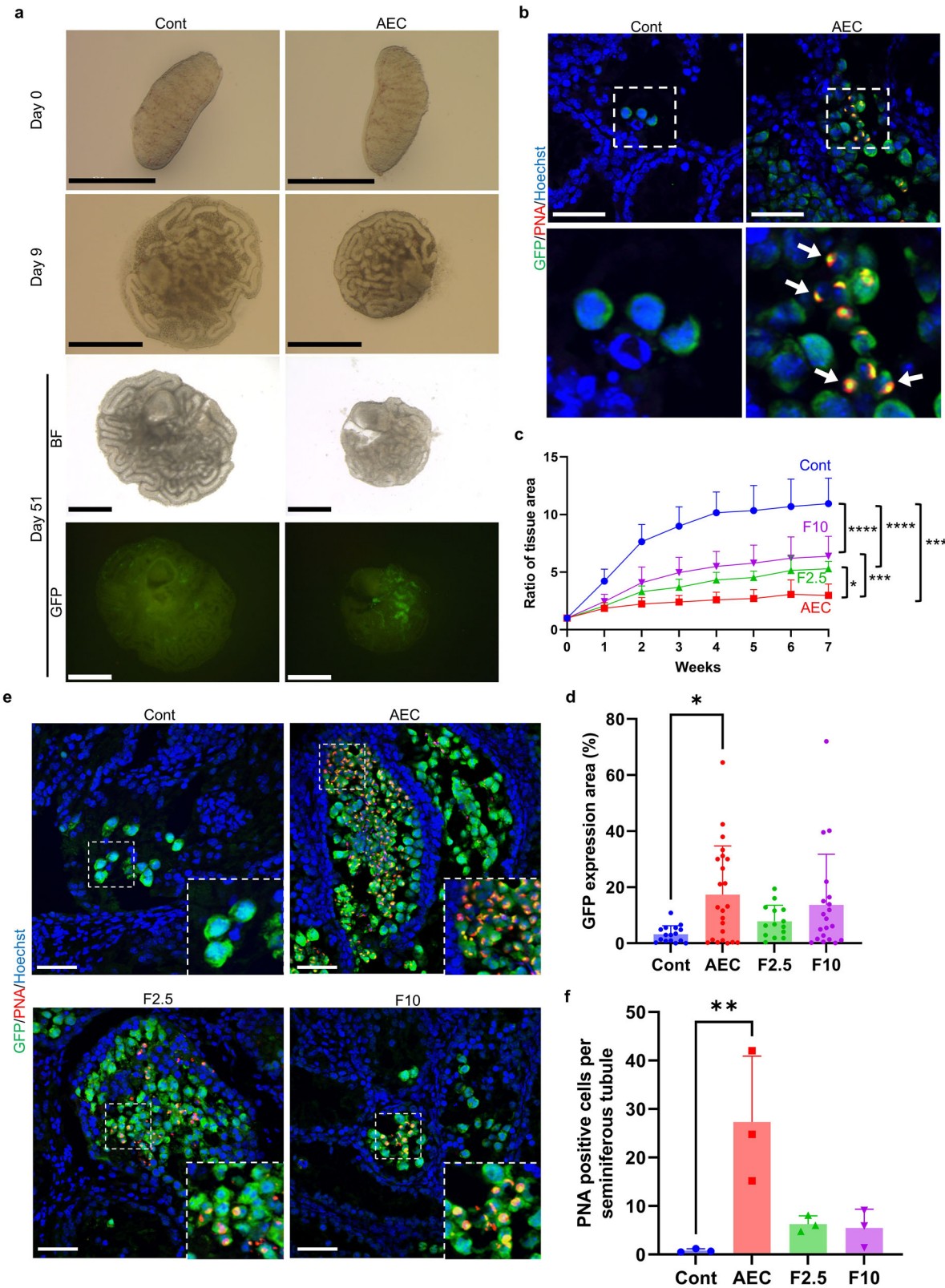

positive cells (Fig. 4c, d and Supplementary Fig. 2b). In contrast, the AEC group at 10% oxygen yielded numerous PNA-positive round and elongated spermatids (Fig. 4c, d and Supplementary Fig. 2c). Furthermore, mechanical dissociation of tissues cultured in AEC at 10% oxygen yielded spermatozoa with flagella (Fig. 4e). These results demonstrate that RTI supplementation under hypoxic conditions enables efficient in vitro spermatogenesis, even from developmentally E12.5 fetal testes.

## Production of fertile offspring using in vitro–generated spermatids from E12.5 fetal testes

Finally, we assessed the fertility of spermatids obtained from cultured E12.5 fetal testes using microinsemination techniques. Round spermatids were collected from testicular tissues cultured in AEC medium at 10% oxygen for 51 days and used for round spermatid injection (ROSI). Elongated spermatids were also present at the time of collection but could not be isolated

**Fig. 3 | In vitro spermatogenesis from E12.5 fetal testes in RTI-supplemented culture conditions. a** Representative stereomicroscope images of testicular tissues derived from E12.5 fetal testes at days 0, 9, and 51. Scale bar = 1 mm. **b** Immunofluorescence analysis of E12.5 testicular tissues cultured for 51 days with AEC-supplemented medium. Sections were stained with anti-GFP antibody (green), PNA (red), and Hoechst 33342 (blue). Enlarged views of boxed regions are shown below. Arrows indicate round spermatids. Scale bar = 50 μm. **c** Growth ratio of testicular tissue during culture. Relative area increase of cultured tissues compared to day 0 is shown. *n* = 6-10. *$p < 0.05$, ***$p < 0.0005$, ****$p < 0.0001$ (Tukey's multiple comparisons test). **d** Quantification of GFP-positive area in E12.5 testicular tissues

after 7 weeks of culture. Data are presented as mean ± s.d. *$p = 0.0219$ (Dunn's multiple comparisons test). **e** Immunofluorescence staining of E12.5 testicular tissues cultured for 51 days with AEC, F2.5, or F10 supplementation. Sections were stained with anti-GFP (green), PNA (red), and Hoechst 33342 (blue). Inset images show enlarged views of boxed regions. Round spermatids were observed in the AEC, F2.5, and F10 groups. Scale bar = 50 μm. **f** Quantification of PNA-positive cells per seminiferous tubule under each culture condition. A significant increase in PNA-positive cells was observed in the AEC supplemented group. **$p = 0.0067$ (Dunn's multiple comparison test).

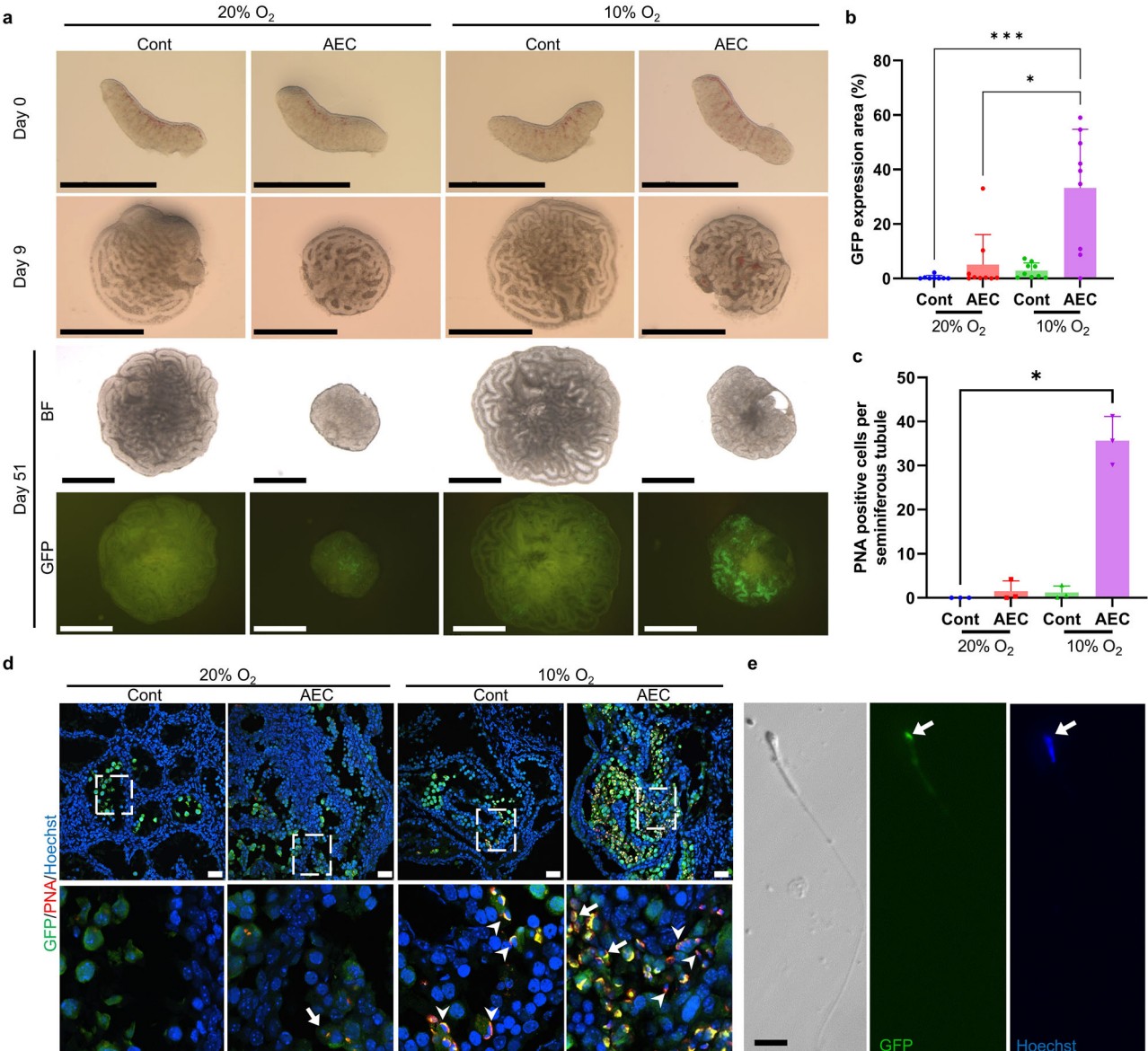

**Fig. 4 | In vitro spermatogenesis from E12.5 fetal testes under hypoxic conditions. a** Representative stereomicroscope images of E12.5 testicular tissues cultured for 51 days under normoxic (20% $O_2$) or hypoxic (10% $O_2$) conditions. Scale bar = 1 mm. **b** Quantification of GFP-positive area at the end of culture (day 50-51). Data are presented as mean ± s.d. *$p = 0.03235$, ***$p < 0.0003$ (Dunn's multiple comparison test). **c** Quantification of PNA-positive cells per seminiferous tubule under hypoxia. Data are presented as mean ± s.d. *$p = 0.0262$ (Dunn's multiple

comparison test). **d** Immunofluorescence staining of E12.5 testicular tissues cultured under various conditions. Sections were labeled with anti-GFP antibody (green), PNA (red), and Hoechst 33342 (blue). Arrows and arrowheads indicate round and elongated spermatids, respectively. Scale bar = 50 μm. **e** Representative image of mechanically dissociated spermatozoa stained with Hoechst 33342 at the end of culture (day 51). Condensed nuclear morphology and acrosome caps labeled by *Acr*-GFP fluorescence were observed in sperm heads (arrows).

for microinsemination due to physical entanglement with surrounding cells (Fig. 5a). A total of 136 oocytes were injected with round spermatids collected from two cultured testicular tissues, resulting in 2 and 13 offspring, respectively (Fig. 5b and Supplementary Table S1). Although 6 offspring

died shortly after birth, this may have been due to maternal neglect, as all were nursed by the same surrogate mother. PCR analysis of ear or tail DNA revealed that 6 of the 9 surviving offspring carried the *Acr*-GFP transgene, consistent with heterozygosity in the cultured testicular tissue (Fig. 5c). To

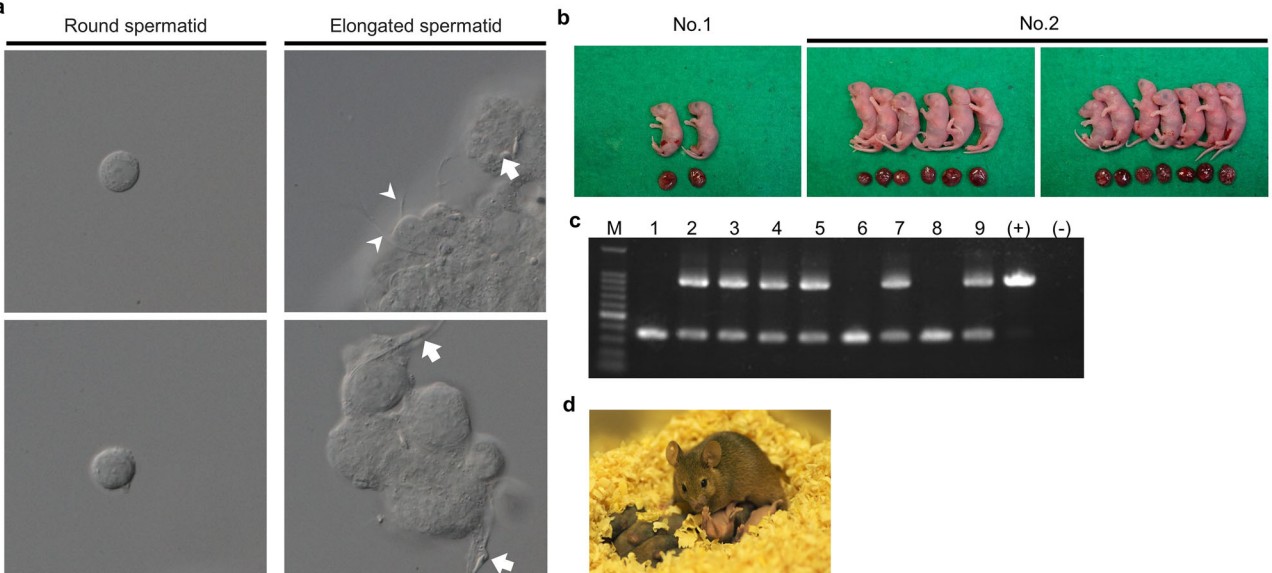

**Fig. 5 | Fertility of in vitro-generated spermatids assessed by microinsemination.** **a** Representative image of round and elongated spermatids observed during microinsemination. Arrows indicate sperm heads, and arrowheads indicate sperm flagella. **b** Photograph of offspring and placentas obtained by round spermatid injection (ROSI). Two and thirteen offspring were obtained from two cultured testicular tissue fragments (No. 1 and No. 2), respectively. **c** Genotyping of offspring by PCR for the *Acr*-GFP transgene. M: molecular size marker (100 bp DNA Ladder); (+) positive control using genomic DNA from homozygous *Acr*-GFP Tg mice; (-) negative control (non-template DNA). The 355 bp band corresponds to the wild-type allele, and the 924 bp band to the transgene allele. **d** Photograph of second-generation offspring obtained by mating mice born via ROSI. These offspring were also healthy and showed normal growth.

confirm their reproductive competence, the surviving F1 offspring were subjected to sibling mating after weaning. We established two breeding pairs, both of which were fertile, producing healthy second-generation (F2) litters of 12 and 7 pups, respectively (Fig. 5d). These litter sizes are within the normal range for this mouse strain, providing definitive evidence that the mice derived from microinsemination were fully fertile

These results demonstrate that culturing E12.5 fetal testes in AEC medium under hypoxic conditions supports complete in vitro spermatogenesis and enables the generation of fertile offspring from fetal testis-derived spermatids, a first for this early developmental stage.

## Expression of retrotransposons in culture

Our studies demonstrated that RTIs improve the efficiency of in vitro spermatogenesis induction in neonatal and early post-sex differentiation fetal testes. RTIs are known to suppress retroviral retrotransposition and inhibit the retrotransposition of the retrotransposon LINE1[11]. Derepression of retrotransposons in male germ cells is associated with spermatogenic arrest and male infertility[12]. Based on these findings, we hypothesized that testicular tissues in culture experience weakened retrotransposon suppression, and that the resulting retrotransposon activation impedes spermatogenesis. We therefore reasoned that the improvement in in vitro spermatogenesis efficiency by RTIs is mediated, at least in part, through suppression of these derepressed retrotransposons. Indeed, Hirano *et al.* reported that neonatal testes cultured for 5 weeks showed significantly higher expression of the retrotransposon LINE1-ORF1p compared with age-matched in vivo testes[13].

To assess whether retrotransposons are expressed under culture conditions, we compared LINE1 expression between in vivo and in vitro systems. We first examined LINE1 ORF1 expression by immunostaining. Neonatal mice at 2.5-7.5 days postpartum were divided into in vivo and in vitro groups, with the former being maintained in vivo and the latter cultured for 5 weeks in control medium. Tissues were fixed at 2 and 5 weeks, and the percentage of seminiferous tubules containing LINE1-positive cells among GFP-positive tubules (indicative of progressing spermatogenesis) was quantified in LINE1-ORF1p–immunostained sections (Fig. 6a, b). In vivo, seminiferous tubules containing ORF1p-positive cells were rare at both

2 and 5 weeks of age (1.85% and 5.12%, respectively). By contrast, ORF1p expression was readily detectable throughout the culture period, with particularly high expression (65.3%) after 2 weeks in vitro. These findings indicate that retrotransposons are expressed at much higher levels in vitro than in vivo.

Next, we examined the expression of LINE1-ORF2 and piRNA-related genes, which are components of the retrotransposon suppression machinery[12], by qPCR. We compared testes collected at E15.5 in vivo with testes cultured in vitro for 3 days (corresponding to the E15.5 developmental stage) in control medium. LINE1 expression, represented by ORF2, was significantly increased under in vitro conditions compared with in vivo (Fig. 6c). Conversely, the expression levels of piRNA-related genes, *Mael*, *Piwil4* and *Tdrd9*, which suppress retrotransposons including LINE1, were significantly decreased in vitro (Fig. 6c).

To investigate whether retrotransposition occurs during organ culture, we next quantified the genomic copy number of LINE1-ORF2. Neonatal littermates were assigned to either an in vivo group or in vitro culture groups, the latter subdivided into a control (RTI-free) group and RTI-treated groups (AZT/AEC). Using genomic qPCR, we analyzed the ORF2 copy number in GFP-positive cells from tissues cultured for a period equivalent to 20–26 days of age, before substantial differences in GFP expression rates emerged. This analysis revealed that ORF2 copy number increased by 1.26-fold in the control group relative to in vivo (Fig. 6d). Conversely, treatment with AZT and AEC reduced the copy number to 0.89- and 0.86-fold, respectively, compared with the control group, although these differences did not reach statistical significance (Fig. 6d). Collectively, these results support the hypothesis that derepression of retrotransposons contributes to impaired in vitro spermatogenesis and that RTIs mitigate this effect.

## Discussion

In this study, we demonstrated that supplementation with RTIs, particularly the combination of AZT, EFV, and ddC (AEC), significantly enhances in vitro spermatogenesis efficiency from mouse testes. Notably, we achieved complete spermatogenesis from E12.5 fetal testes, a developmental stage previously considered too immature to support in vitro germ cell

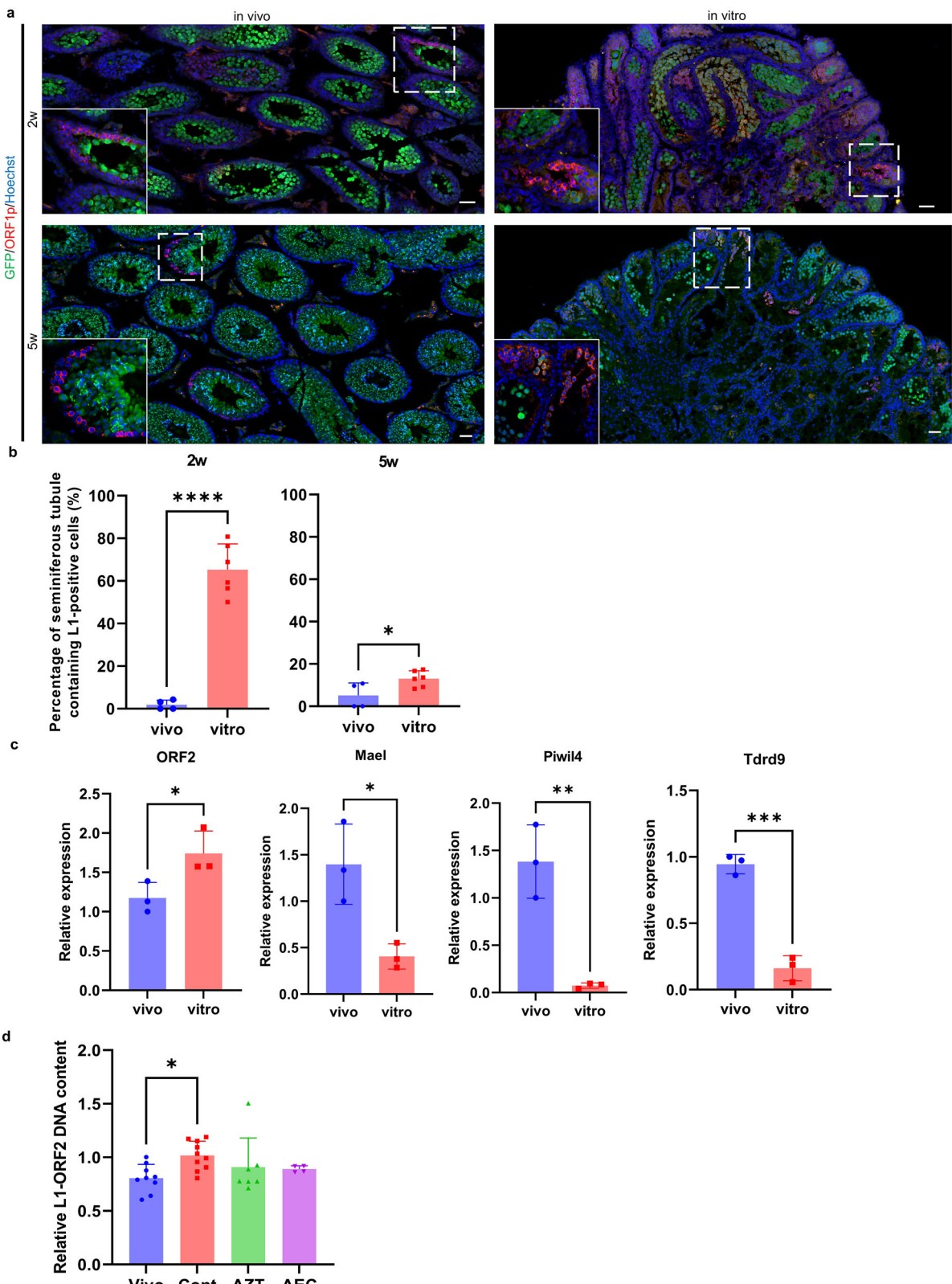

**Fig. 6 | Overexpression of retrotransposons in cultured testicular tissues. a** LINE1 expression in cultured neonatal mouse testes. Testicular tissues from in vitro cultured 4.5-day-old neonatal testes and in vivo testes of the same age were labeled with anti-GFP antibody (green) and anti-LINE1-ORF1p antibody (red), and nuclei were stained with Hoechst 33342 dye (blue). Scale bar = 50 μm. **b** Percentage of semi-niferous tubules containing LINE1-positive germ cells in in vivo and in vitro *Acr*-GFP mouse testes. *p = 0.0314, ****p < 0.0001 (Student's t test). **c** Expression of *Piwi*-related genes (*Mael*, *Piwil4*, *Tdrd9*) and *LINE1-ORF1* in cultured fetal testes.

Gene expression in E12.5 fetal testes cultured to the equivalent E15.5 stage (in vitro) and in vivo testes from E15.5 embryos was examined by quantitative PCR. Expression was normalized to the germ cell-specific gene *Mvh*. *p < 0.05 (*ORF1*: p = 0.0473; *Mael*: p = 0.0192), **p = 0.0043, ****p = 0.0003 (Student's t test). **d** Relative LINE1-ORF2 DNA content in germ cells from in vivo controls and various culture conditions. The values were normalized to 5S ribosomal RNA gene and presented as mean ± s.d. *p = 0.0291 (Dunn's multiple comparison test).

differentiation[4]. The present work therefore extends the applicability of testis organ culture and provides a novel platform to investigate the earliest stages of spermatogenesis in a controlled in vitro environment. The successful induction of complete spermatogenesis from E12.5 testes demonstrates that the testicular primordium at this stage contains all the necessary precursor cells to autonomously assemble the required architecture and support the entire process in vitro, independent of systemic cues from other organs such as the pituitary gland.

RTIs are known to inhibit endogenous retroelement activity[11,14,15], particularly that of LINE1, whose expression is readily activated in fetal germ cells and has been implicated in genomic instability and cell death[16]. Suppressing such activity may protect developing germ cells and preserve their viability in culture. Previous studies have reported upregulation of retrotransposons during in vitro culture of neonatal and fetal testes, suggesting that culture-associated stress can trigger their derepression[13,17]. Our present findings support this notion. Specifically, we observed downregulation of several piRNA-related genes, upregulation of LINE1 expression, and a corresponding increase in LINE1 genomic copy number in cultured testicular tissues. By mitigating this stress response, RTIs may create a more permissive environment for meiotic progression.

We also found that hypoxic culture conditions (10% $O_2$) substantially improved spermatogenesis efficiency compared to normoxia (20% $O_2$), consistent with our earlier reports in neonatal and rat testis cultures[8,9]. Hypoxia may reduce oxidative stress and improve cellular energy metabolism in fetal tissues, or more closely mimic the in vivo fetal environment. Interestingly, hypoxia alone had limited benefit in control medium, but synergized strongly with AEC, suggesting that these two interventions act through distinct cellular pathways. This synergy warrants further molecular characterization.

Although RTI supplementation clearly enhances germ cell differentiation, it was also associated with reduced overall tissue growth. This suggests that AEC impairs the proliferation of somatic or germ cells, possibly by interfering with nucleotide metabolism or cell cycle progression. Our attempt to stimulate tissue growth by adding FBS during the early culture phase increased testis size but did not improve spermatogenic efficiency. Other agents capable of promoting testis tissue growth likely exist and could help mitigate the adverse effects of AEC. Identifying such agents, as well as clarifying the precise timing and cell types affected by RTIs, will be important for further refining the culture conditions.

The most notable finding in this study is the production of fertile offspring from in vitro–generated spermatids originating from E12.5 fetal testes. Previous studies have achieved offspring from neonatal testis–derived spermatids, but this is, to our knowledge, the earliest stage of origin yet demonstrated to yield functional haploid cells in vitro. The fertility of the F1 generation was confirmed by successful sibling matings, which produced grossly normal and reproductively competent F2 litters of normal size, providing definitive functional proof. This not only provides a powerful system for dissecting the molecular events of early spermatogenesis but also suggests that even prepubertal or fetal germ cell populations may hold regenerative potential if placed in an optimized environment.

Future studies should focus on the transcriptional and epigenetic states of germ cells and supporting somatic cells under RTI and hypoxic conditions. Single-cell RNA-seq and chromatin accessibility profiling could elucidate how RTIs reshape the differentiation landscape. Additionally, the reconstitution of germ cell development from stem cell–derived fetal testis–like tissue may now be informed by these findings, offering a step toward modeling human germline development in vitro. To ensure the translational relevance of these advances, future work should also include thorough health assessments of the offspring, such as behavioral testing, organ histopathology, and transgenerational epigenetic profiling.

In conclusion, this study establishes that complete and functional spermatogenesis can be achieved from as early as E12.5 mouse fetal testes using RTI-supplemented, hypoxia-conditioned organ culture. These findings deepen our understanding of early testicular development and open new avenues for reproductive biology and regenerative medicine.

## Methods

### Animals

*Acr*-GFP Tg mice (strain: ICR, C57BL/6 and their mixture) supplied by Riken BRC were used. The spermatogenic germ cells of this mouse express GFP at the middle-pachytene stage onward[18]. Homologous *Acr*-GFP Tg mice (8-24-week-old) were cohoused overnight with 8-23-week-old ICR (CLEA Japan Inc., Tokyo, Japan) female mice and mated. The day on which a vaginal plug was identified in the mother was E0.5, and the fetal age was determined. Animals were fed *ad libitum* with MF hard pellets (Oriental Yeast Co., Ltd., Tokyo, Japan). Drinking water was acidified to pH 2.8-3.0 by HCl. Neonatal mice were euthanized by decapitation following anesthesia by hypothermia on ice. Pregnant mice were euthanized by an intraperitoneal injection of an overdose of an anesthetic mixture (0.3 mg/kg medetomidine, 4.0 mg/kg midazolam, and 5.0 mg/kg butorphanol). All animal experiments conformed to the Guide for the Care and Use of Laboratory Animals and were approved by the Institutional Committee of Laboratory Animal Experimentation (Animal Research Center of Yokohama City University; protocol no. F-A-23-023). We have complied with all relevant ethical regulations for animal use.

### Organ culture method

For neonatal mouse (1.5-5.5dpp) testis culture, the testes of the pup mice were removed and decapsulated. These testes were sectioned into 4-8 pieces and used for culture. Fetal mouse testes (E12.5 to E14.5) were cultured as whole without sectioning, carefully removing the mesonephros attached to the removed testes. The culture medium and culture method were gas-liquid phase interface culture, as previously described, in which testicular tissue was placed on an agarose gel with the lower half immersed in aMEM supplemented with 10% Knockout Serum Replacement (KSR, Thermo Fisher Scientific, Inc.). The RTIs used were AZT (FUJIFILM Wako Pure Chemical, Japan, Cat#015-14704), d4T (FUJIFILM Wako Pure Chemical, 197-15871), ddC (Abcam, Cat#ab142240), EFV (Sigma-Aldrich, Cat#SML0536), and ADV (Abcam, Cat#ab143063). Incubators were maintained at 34 °C with air (20% $O_2$) or 10% $O_2$ of 5% $CO_2$. The medium was changed once a week.

### Observations and immunohistochemistry

Cultured tissues were observed weekly with a stereo fluorescence microscope (Leica M205FA; Leica Camera AG, Wetzlar, Germany) and bright-field and GFP fluorescence images were captured.

Testicular tissue was immersed in 4% PFA, fixed overnight at 4 °C, and embedded with O.C.T. Compound (Sakura Finetek Japan, Japan). Cryosections were made with a cryostat at 7 μm thickness and used for fluorescent immunostaining. Fluorescent immunostaining was performed as described in a previous paper. The following antibodies were used as the primary antibody: chicken anti-GFP (1:1000, Abcam, Cat#ab13970) and rabbit anti-LINE1-ORF1p (1:500, Abcam, Cat#ab216324) antibodies. The secondary antibodies used were goat anti-chicken IgY, conjugated to Alexa Fluor 488 (1:200; Thermo Fisher Scientific, Inc., Cat#A-11039) and goat anti-rabbit IgG conjugated to Alexa Fluor 555 (1:200; Thermo Fisher Scientific, Inc., Cat#A-21428). Nuclei were stained with Hoechst 33342 dye and acrosomes of haploid cells with Lectin PNA From Arachis hypogaea (peanut) conjugated to Alexa Flour 568 (1:1000; Thermo Fisher Scientific, Inc., Cat#L32458). Specimens were observed under a microscope (Keyence BZ-X800) or confocal laser microscope (Nikon AX).

### Measurement of *Acr*-GFP expression area

Brightfield and GFP fluorescence images of cultured testicular tissues were captured using a DFC7000 color camera (Leica, Wetzlar, Germany) attached to an M205 fluorescence stereomicroscope, and the *Acr*-GFP expression area was calculated as previously described[19]. The total area of testicular tissue was measured by tracing the outline of the tissue using the freehand selection tool in National Institutes of Health ImageJ (version 1.54). The GFP expression area was measured by automatically selecting the

green color of GFP on fluorescence photographs using the color threshold function in ImageJ.

## Flow cytometric analysis

Flow cytometric analysis was performed according to previous reports[20]. Cultured tissues were digested with 2 mg/mL collagenase in PBS for 20 min at 37 °C, then followed by 0.25% trypsin for 10 min at 37 °C. The reaction was stopped by adding Dulbecco's modified Eagle's medium containing 10% FBS, then the cells were filtered through a 40 μm pore size cell strainer (Becton Dickinson, Franklin Lakes, NJ) was used to filter the cells, which were pelleted by centrifugation (220 g, 5 min) and resuspended in PBS containing 3% (vol/vol) FBS. Cells were fixed in 2% PFA in PBS for 20 minutes on ice and washed twice with PBS. Cells were then treated with 70% ethanol and incubated overnight at 4 °C. After centrifugation (2200 g, 5 min), the cell pellet was washed twice with 0.2% PBST [PBS containing 0.2% Triton X-100] containing 10% FBS. The cells were then stained with Alexa Fluor 488-conjugated polyclonal GFP antibody (1:300; Thermo Fisher Scientific, Cat#A-21311) diluted in 0.2% PBST containing 5% BSA at 37 °C for 1 h. The cells were washed twice with 0.2% PBST + 10% FBS. The supernatant was then removed and stained with 300 μL PI staining solution [20 μg/mL PI stock (P4864-10ML; Merck), 200 μg/mL RNase A (Nacalai Tesque, Kyoto, Japan) in 0.2% PBST] at 37 °C for 15 minutes. The samples were applied to a Guava easyCyte flow cytometer (Merck, Darmstadt, Germany). Details of the gating and sorting strategy are provided in Supplementary Fig. 3.

## Round spermatid injection

Round spermatid injection was performed as previously described[21]. Cultured testicular tissues were dissected under a stereomicroscope. Mature MII oocytes were collected from superovulated 9-12-week-old B6D2 females and freed from cumulus cells by treatment with 0.5% bovine testicular hyaluronidase. The oocytes were then preactivated with $Ca^{2+}$-free CZB medium containing 2.5 mM $SrCl_2$. Round spermatids were subsequently collected and injected into these preactivated oocytes at Telophase II (40-50 minutes after activation) using a piezo-driven micromanipulator. Fertilized oocytes were cultured for 24 hours, and the resulting two-cell embryos were transferred into the oviducts of pseudopregnant 9-12-week-old ICR females. A live fetus recovered on day 19.5 was reared by a nursing ICR foster mother.

## Genotyping

Genomic DNA from offspring was extracted by incubation of tail fragments or ear pieces in 50 mM NaOH at 94 °C for 20 min, followed by neutralization with Tris-HCl (pH 8.0). The primer sequences used for genotyping *Acr*-GFP Tg are shown in Supplementary Table S2. A band of 355 bp and a band of 924 bp are detected for wild-type and Tg mice, respectively.

## Quantitative RT-PCR analysis

Fetal testes cultured to the equivalent of E15.5 and E15.5 fetal mouse testes were collected and homogenized, respectively, and RNA was extracted using an RNA purification kit (RNeasy Mini Kit, QIAGEN). Reverse transcription was performed using a reverse transcription reaction kit (ReverseTra Ace qPCR RT Master Mix with gDNA Remover; Toyobo). qPCR analysis was performed with Power SYBR Green Master Mix (Applied Biosystems, Foster City, CA, USA) on a StepOne Plus real-time qPCR system (Applied Biosystems). Gene expression levels were calculated as deltaCt normalized by the mean Ct value of *Mvh*. The primers used in this study are shown in Supplementary Table S2.

## Quantitative genomic PCR for determination of LINE1 copy number

Testicular tissue fragments cultured to the equivalent of 20–26 days of age, along with testes from their in vivo littermate controls, were incubated in 2 mg/mL collagenase and treated for 15 min at 37 °C. After centrifugation

and removal of the supernatant, the pellet was treated with 0.25% trypsin for 10 min at 37 °C. The reaction was stopped by adding culture medium, and the cell suspension was filtered to remove debris. Following another centrifugation and supernatant removal, the cells were resuspended in PBS with 3% FBS to a final concentration of $1–5 \times 10^6$ cells/mL. Dead cells were labeled by adding PI.

GFP-positive cells were then isolated using a cell sorter (MoFlo Astrios; Beckman Coulter). The sorted cells were centrifuged, the supernatant was removed, and genomic DNA was extracted using the QIAamp DNA Micro Kit (QIAGEN). This DNA was used as a template for qPCR. The copy number of the ORF2 region of the LINE1 retrotransposon was quantified, and the non-mobile 5S ribosomal RNA sequence was used as a control for normalization[22]. The relative genomic ORF2 content was normalized to the 5S genomic content. qPCR analysis was performed with Power SYBR Green Master Mix on a StepOne Plus real-time qPCR system.

## Statistics and reproducibility

Statistical analyses were performed using GraphPad Prism v.9. Data are presented as mean ± s.d. The sample size ($n$) is represented by individual dots in the graphs, except for Fig. 1e ($n = 6$ independent organ culture samples) and Fig. 3c ($n = 6–10$ independent testes), which are line graphs. For neonatal testes, $n$ represents independent organ culture samples (testis tissue fragments) distributed across experimental groups. For fetal testes, n represents biologically independent testes. Statistical tests used are described in the figure legends. P-values are provided in the figures or legends where possible, with $p < 0.05$ considered significant. Qualitative data, such as microscopy images, were confirmed by at least three independent observations using testes from different biological individuals with similar results.

## Reporting summary

Further information on research design is available in the Nature Portfolio Reporting Summary linked to this article.

## Data availability

Source data underlying the graphs and charts in the main figures are provided in Supplementary Data 1. The unedited electrophoresis image is included in Supplementary Fig. 4. All other data supporting the findings of this study are available from the corresponding author upon reasonable request.

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

## Acknowledgements
We would like to thank the members of the Department of Regenerative Medicine for their experimental assistance and discussion. We thank Dr. Alex Bortvin of Carnegie Institution for Science for kindly providing the anti-LINE1-ORF1p antibody, which was essential for the initial stages of this study. This work was supported by KAKENHI grants-in-aid from MEXT, Japan (24H02055 to Y.I.-Y.; 23H04956 to K.I.; 19H05758 to A.O.; 18H05546 and 22H00485 to T.O.; 20K21657, 24K21284 and 25K02455 to T.S.); by an AMED (JP24gn0110086 to T.O.; JP24mk0121304 to T.S.); by JST CREST (JPMJCR21N1 to T.O.) and by a Grant for Strategic Research Promotion of Yokohama City University (SK202403 to T.S.).

## Author contributions
M.N., Y.O.-S., S.K. and T.S. conceived, designed, and conducted the experiments, and performed data analyses; M.N., Y.O.-S., S.K., Y.I.-Y., T.M., M.K. and T.S. performed organ culture experiments; S.M., K.I., N.O. and A.O. performed round spermatid injection experiments; M.N., T.O. and T.S. wrote the manuscript, incorporating feedback from all the authors.

## Competing interests
The authors declare no competing interests.
