## [Transparent Peer Review file · Communications Biology]

Reverse transcriptase inhibitors enable the generation of fertile spermatids from fetal mouse testes *in vitro*

Corresponding Author: Dr Takuya Sato

Version 0:

Reviewer comments:

Reviewer #1

(Remarks to the Author)

This study establishes a groundbreaking organ culture system enabling complete spermatogenesis from E12.5 mouse fetal testes—a developmental stage previously considered too immature for *in vitro* germ cell differentiation. By optimizing culture conditions with reverse transcriptase inhibitors (RTIs, specifically the AEC combination) and hypoxic (10% O₂) environments, the authors achieved functional spermatid formation and produced healthy, fertile offspring via microinsemination. This work advances reproductive biology by providing a novel platform to study early spermatogenesis and demonstrates the regenerative potential of fetal germ cells under optimized conditions. Major Revisions Required Prior to Acceptance While the findings are transformative, significant revisions are needed to strengthen mechanistic insights.

1. The authors hypothesize that RTIs suppress retroelement activity (e.g., LINE-1) to protect germ cells, but direct evidence (e.g., RNA-seq, LINE-1 expression analysis, or DNA damage markers) is lacking. Can the authors provide molecular data (e.g., single-cell RNA-seq or chromatin profiling) to confirm that RTIs mitigate retrotransposon-driven genomic instability in cultured fetal testes?
2. While hypoxia improved spermatogenesis, the rationale for choosing 10% O₂ (vs. other levels) and its interaction with RTI supplementation remains unclear. Was hypoxia titrated systematically (e.g., 5%, 10%, 20% O₂), and how do oxygen levels affect somatic cell function (e.g., Sertoli cell metabolism or vascularization) in the culture system?
3. The study highlights E12.5 as a breakthrough stage, but comparisons to later stages (e.g., E14.5) are limited. Why does AEC fail to rescue growth in E12.5 cultures despite improving spermatogenesis? Could somatic cell immaturity at E12.5 limit proliferative capacity, and how might this be decoupled from differentiation?
4. While fertile offspring were generated, detailed health assessments (e.g., behavioral tests, organ histopathology, or transgenerational epigenetic analysis) are absent. Are there subtler phenotypic or epigenetic abnormalities in microinsemination-derived offspring that might reflect incomplete reprogramming of fetal germ cells?
5. While the production of fertile offspring is a major achievement and confirms functional competence, a more detailed assessment of the health of the F1 generation derived from E12.5 *in vitro* spermatids would strengthen the claim of 'healthy' offspring. Were there any gross morphological abnormalities, significant differences in body/organ weights, or signs of premature aging observed? At minimum, data on litter sizes and viability of offspring from the sibling matings (F2 generation) should be included to assess reproductive fitness beyond the F1.

Reviewer #2

(Remarks to the Author)

Nishida et al. reported a significant advancement in *in vitro* spermatogenesis using fetal mouse testes at E12.5 to obtain functional spermatids capable of producing offspring. The authors attributed this advancement primarily to their use of reverse transcriptase inhibitors (RTIs). This study represents a breakthrough in *in vitro* spermatogenesis starting from testes at such an early stage, and the combination of RTIs with hypoxia is novel.

The key weakness is the relatively low efficiency of differentiation, and the mechanism of RTI action remains unclear. The authors suggested that LINE-1 expression might increase during meiosis and be detrimental to spermatogenesis, proposing that RTIs counteract this effect. However, no experimental evidence supports this hypothesis in the study. Additionally, as RTIs may impair cell growth and differentiation, it would be valuable to assess their toxicity on gonadal development. Another unresolved question is whether RTIs can penetrate the seminiferous tubules in the *in vitro* system, given that the

blood-testis barrier was still intact and functional. Do RTIs act directly on germ cells or indirectly via somatic cells? As the authors noted, single-cell RNA-seq could elucidate how RTIs reshape differentiation. Had such data been included, the mechanistic insights would significantly strengthen the study.

Reviewer #3

(Remarks to the Author)

The manuscript by Nishida et al. reports the production of fertile spermatids from fetal mouse testes in culture. This study provides potentially interesting results that would help future attempts to generate sperm from embryonic stem cells. However, the results are largely preliminary and need further studies to make their conclusions more convincing.

Major comments

1. The evidence for "spermatogenesis" is based only on the expression and relative area of Acr-GFP or PNA. An increase in the GFP-positive area can be attributed to proliferation of germ cells, facilitated spermatogenic progression, or loss of GFP-negative cell populations. Reference #16 was given to justify the experimental method, but it was published in 1999 from a separate laboratory. More vigorous evaluation of spermatogenesis, such as morphology with high resolution, is required. Generation of fertile spermatids supports the occurrence of complete spermatogenesis, but only two seminiferous tubules were tested. No control was provided to assess whether the success rates of 6-12% were low or high. The conclusion such as "RTI supplementation improves the efficiency of in vitro spermatogenesis in fetal testes" (Lines 127-128) is not well supported.
2. As the authors pointed out, "The mechanism by which RTIs promote spermatogenesis remains unclear" (Line 211) is a major weakness of this paper. Some potential mechanisms, such as inhibition of LINE1, can be easily tested. The authors can also extend the discussions based on the dosage, specificity, and effective time of RTIs. Sertoli cell differentiation can be monitored by expression of a couple of key molecules such as SOX9 and DMRT1.
3. The culture conditions need to be better justified. "Fetal Bovine Serum (FBS), a common culture supplement" (Lines 154-155) is not true. 10% FBS is known to promote the growth of fibroblast but not of epithelial cells. Furthermore, the basic medium already contains serum replacement supplements. Adding FBS can be toxic. Did the authors adjust the basic medium with FBS supplementation? The culture was performed at 34°C. This is not the natural environment for fetuses. Did the authors optimize the temperature for the culture of fetal testes?
4. Discussion "The fact that spermatogenesis could be induced from E12.5 testes --- suggests that the minimal testicular architecture and somatic support required for germ cell development are already in place at this stage" (Lines 221-223) is ambiguous. All cells there underwent differentiation during culture. This experimental design revealed that all precursor cells are there to support spermatogenesis independent of other parts of the body such as the pituitary.
5. Figure 3F: "Quantitation of PNA-positive cells per seminiferous tubule" does not provide correct measurements. Some seminiferous tubules in a section are round and small while others are long. Quantitation must be against a consistent unit size.

Minor comments

- The title needs rephrasing.
- Line 38: "sperm" should be "spermatozoa."
- Line 54: "also reported in those contexts." --- The meaning is unclear.
- Line 59: "decreasing developmental stage" is awkward. "earlier developmental stages"?
- Line 271: Explain why "Drinking water was acidified."
- Line 298: "goat anti-chicken IgG" --- Chicken does not have IgG.
- Lines 347-348: Give the name of the target gene, Acr-GFP, for PCR genotyping in the text. In Table S2, specify the strand and orientation targeted by each primer and the combination of primers for amplifying WT and TG bands.

Version 1:

Reviewer comments:

Reviewer #1

(Remarks to the Author)

The authors have addressed my questions. Thanks for their hard work.

Reviewer #3

(Remarks to the Author)

The revised manuscript by Nishida et al. has improved considerably. The authors have properly addressed most of my previous concerns. I have suggestions only for minor changes.

Lines 69 & 181: "immature E12.5 fetal testes" --- The meaning of "immature" is unclear here. It is better to remove it.

Lines 451 & 483: “cultured with --- medium” should read better “cultured in --- medium.”

We would like to sincerely thank the reviewers for their constructive comments, which we have used as the basis for revising our manuscript.

Reviewers' comments:

Reviewer #1 (Remarks to the Author):

This study establishes a groundbreaking organ culture system enabling complete spermatogenesis from E12.5 mouse fetal testes—a developmental stage previously considered too immature for *in vitro* germ cell differentiation. By optimizing culture conditions with reverse transcriptase inhibitors (RTIs, specifically the AEC combination) and hypoxic (10% O₂) environments, the authors achieved functional spermatid formation and produced healthy, fertile offspring via microinsemination. This work advances reproductive biology by providing a novel platform to study early spermatogenesis and demonstrates the regenerative potential of fetal germ cells under optimized conditions. Major Revisions Required Prior to Acceptance While the findings are transformative, significant revisions are needed to strengthen mechanistic insights.

1. The authors hypothesize that RTIs suppress retroelement activity (e.g., LINE-1) to protect germ cells, but direct evidence (e.g., RNA-seq, LINE-1 expression analysis, or DNA damage markers) is lacking. Can the authors provide molecular data (e.g., single-cell RNA-seq or chromatin profiling) to confirm that RTIs mitigate retrotransposon-driven genomic instability in cultured fetal testes?

Reply: This is a critical point, and we thank the reviewer for highlighting the need for direct molecular evidence. To address this, we have performed several new experiments to directly investigate LINE-1 activity and the effect of RTIs in our organ culture system.

First, we confirmed the premise of our hypothesis that LINE-1 activity is indeed elevated in our culture system. As shown in our new data, both LINE-1 protein and transcript levels were significantly upregulated in cultured testes without RTIs, as measured by immunostaining and RT-qPCR, respectively (New Fig. 6a-c). Mechanistically, we also found that the expression of several piRNA-pathway genes, which endogenously suppress retrotransposons, was downregulated during culture.

Next, we tested whether this LINE-1 activation leads to genomic instability and if RTIs can counteract this effect. We quantified the LINE-1 copy number in genomic DNA using qPCR. We observed an increase in the LINE-1 copy number in germ cells from cultured testes. Most importantly, this increase showed a tendency to be suppressed in the RTI-treated group (New Fig. 6d).

Together, we believe this new set of data provides direct molecular evidence to support our hypothesis that RTIs protect germ cells by suppressing LINE-1-mediated retrotransposition and mitigating genomic instability in cultured fetal testes. We are grateful for this suggestion, as these experiments have substantially strengthened the conclusions of our manuscript.

2. While hypoxia improved spermatogenesis, the rationale for choosing 10% O₂ (vs. other levels) and its interaction with RTI supplementation remains unclear. Was hypoxia titrated systematically (e.g., 5%, 10%, 20% O₂), and how do oxygen levels affect somatic cell function (e.g., Sertoli cell metabolism or vascularization) in the culture system?

Reply: Thank you for this important question regarding the hypoxic culture conditions. We would like to explain the rationale for our choice and our view on the underlying mechanism.

The choice of 10% O₂ was based on a previous study from our laboratory (Feng et al., 2023 (ref. 10)), in which we systematically determined this concentration to be optimal for supporting spermatogenesis while maintaining tissue integrity in this culture system. We therefore applied this established condition for the current study.

The question of how different oxygen levels specifically affect somatic cell function is an important one; however, a detailed mechanistic investigation is beyond the scope of our present work. In our view, normal atmospheric oxygen concentration (approx. 20%) can cause excessive oxidative stress in *in vitro* organ culture systems, potentially damaging various cells. We believe that the 10% oxygen environment alleviates this and contributes to maintaining the overall homeostasis of the testicular tissue.

A detailed verification of this hypothesis remains an important task for future research.

3. The study highlights E12.5 as a breakthrough stage, but comparisons to later stages (e.g., E14.5) are limited. Why does AEC fail to rescue growth in E12.5 cultures despite improving spermatogenesis? Could somatic cell immaturity at E12.5 limit proliferative capacity, and how might this be decoupled from differentiation?

Reply:

Thank you for these insightful questions regarding our focus on the E12.5 stage and the effects of our treatment.

First, regarding our focus on E12.5, as you noted, direct comparisons to later fetal stages in this study are limited. In our previous work (ref. 4), we examined multiple fetal stages and found that testes earlier than E14.5 were particularly difficult to culture to the point of complete spermatogenesis. In fact, we were unable to induce spermatogenesis from E13.5 or earlier stages. In the present study, we again included E14.5 testes to confirm that AEC also enhances *in vitro* spermatogenesis (Fig. 2). We therefore concentrated on E12.5, as achieving complete spermatogenesis from such an early, poorly differentiated testis has long been a major technical barrier.

Your question about how RTI treatment improves spermatogenesis while failing to promote growth, effectively 'decoupling' proliferation from differentiation, is an excellent and astute observation.

We interpret this phenomenon as a result of two distinct pharmacological effects of the RTIs:

- An off-target effect on cell proliferation: The growth inhibition may be a side effect. Certain RTIs are known to cause mitochondrial toxicity by being incorporated by mitochondrial DNA polymerase (doi:10.3923/ijp.2006.152.162). This off-target effect could plausibly limit the proliferative capacity of the rapidly dividing somatic cells, which, as you suggest, may be particularly vulnerable at the immature E12.5 stage.
- An on-target effect on differentiation: In contrast, the improvement in spermatogenesis is the intended effect, driven by the suppression of retrotransposon activity, which protects germ cells and facilitates their differentiation.

Therefore, we consider that the apparent decoupling you noted is due to these two separate mechanisms: an off-target anti-proliferative effect and an on-target pro-differentiation effect.

4. While fertile offspring were generated, detailed health assessments (e.g., behavioral tests, organ histopathology, or transgenerational epigenetic analysis) are absent. Are there subtler phenotypic or epigenetic abnormalities in microinsemination-derived offspring that might reflect incomplete reprogramming of fetal germ cells?

Reply: Thank you for raising this excellent and critical point. The possibility of subtle phenotypic or epigenetic abnormalities in subsequent generations is indeed one of the most important considerations for the long-term safety of our technology.

While we fully acknowledge that such a comprehensive transgenerational analysis is a vital future direction, the primary scope of the current study was to first establish the fundamental proof-of-principle: whether our culture system could produce functionally competent spermatids capable of yielding healthy and fertile F1 offspring.

In light of your comment, we will add a paragraph to the Discussion section to address this point explicitly (Line 306-309).

5. While the production of fertile offspring is a major achievement and confirms functional competence, a more detailed assessment of the health of the F1 generation derived from E12.5 *in vitro* spermatids would strengthen the claim of 'healthy' offspring. Were there any gross morphological abnormalities, significant differences in body/organ weights, or signs of premature aging observed? At minimum, data on litter sizes and viability of offspring from the sibling matings (F2 generation) should be included to assess reproductive fitness beyond the F1.

Reply: Thank you for your insightful comment. We fully agree that a more detailed assessment of the health and reproductive fitness of subsequent generations is a crucial point for strengthening the claims of our study. To that end, we assessed the reproductive competence of the F1 generation. Specifically,

we successfully mated two F1 sibling pairs, which produced litters of 12 and 7 pups (F2 generation), respectively. These litter sizes are well within the normal range for this strain of mice.

This information directly addresses your request for 'data on litter sizes' and provides strong evidence for the normal reproductive fitness of the F1 generation derived from our *in vitro*-produced spermatids. While we acknowledge that this is a limited number of breeding pairs, the normal litter sizes and the gross health of the F2 generation provide significant support for the robustness and safety of our method. To incorporate this clarification and address your concerns, we revised the manuscript as follows:

- Results section: We explicitly state that two F1 sibling pairs were mated and produced healthy litters of 12 and 7 pups, respectively, and that the F2 animals were grossly normal (Line 196-201).
- Discussion section: We revised our claim from 'healthy' to 'grossly normal and reproductively competent' (Line 297). We also added that the normal F2 litter sizes further support the functional competence of the gametes generated via our *in vitro* method.

We believe these additional data and revisions fully address your concerns. We thank you again for this valuable feedback.

Reviewer #2 (Remarks to the Author):

Nishida et al. reported a significant advancement in *in vitro* spermatogenesis using fetal mouse testes at E12.5 to obtain functional spermatids capable of producing offspring. The authors attributed this advancement primarily to their use of reverse transcriptase inhibitors (RTIs). This study represents a breakthrough in *in vitro* spermatogenesis starting from testes at such an early stage, and the combination of RTIs with hypoxia is novel.

The key weakness is the relatively low efficiency of differentiation, and the mechanism of RTI action remains unclear. The authors suggested that LINE-1 expression might increase during meiosis and be detrimental to spermatogenesis, proposing that RTIs counteract this effect. However, no experimental evidence supports this hypothesis in the study. Additionally, as RTIs may impair cell growth and differentiation, it would be valuable to assess their toxicity on gonadal development. Another unresolved question is whether RTIs can penetrate the seminiferous tubules in the *in vitro* system, given that the blood-testis barrier was still intact and functional. Do RTIs act directly on germ cells or indirectly via somatic cells? As the authors noted, single-cell RNA-seq could elucidate how RTIs reshape differentiation. Had such data been included, the mechanistic insights would significantly strengthen the study.

Reply: Thank you for your comprehensive and insightful comments regarding the limitations and

future directions of our study. We would like to address each of the points you raised.

- Regarding Differentiation Efficiency: We agree that improving the differentiation efficiency remains an important future goal. Nonetheless, we consider the reproducible induction of functional spermatids from the E12.5 testis—a developmental stage from which successful spermatogenesis has been extremely challenging—to be a significant advance in the field.
- Regarding Evidence for the RTI Mechanism: To address this central issue, we have added new data that directly demonstrates the activation of LINE-1 and its suppression by RTIs (New Fig. 6). As detailed in our response to Reviewer #1, Comment 1, these results strongly support our hypothesis.
- Regarding RTI Toxicity Assessment: Regarding the potential cytotoxicity of RTIs, as discussed in our response to Reviewer #1, Comment 3, we recognize that the inhibition of tissue growth is likely an off-target effect (e.g., mitochondrial toxicity). It is important to distinguish this side effect from the on-target pro-differentiation effect achieved by suppressing retrotransposons.
- Regarding Blood-Testis Barrier (BTB) Permeability: Your question about BTB permeability is fundamental. In mice, the BTB is generally established around postnatal day 14. Therefore, testes initiated from E12.5 and cultured in our system would remain without a functional BTB for more than 22 days, allowing RTIs to reach germ cells directly.
- Regarding Future Directions and scRNA-seq: To resolve remaining questions, including whether RTIs act directly on germ cells or indirectly via somatic cells, we agree that scRNA-seq is an extremely effective next step, as you suggested. We plan to use this technology in the future to help elucidate these mechanisms and develop more efficient culture methods.

We sincerely appreciate your thoughtful critique, which has both strengthened the current manuscript and helped shape the future direction of our research.

Reviewer #3 (Remarks to the Author):

The manuscript by Nishida et al. reports the production of fertile spermatids from fetal mouse testes in culture. This study provides potentially interesting results that would help future attempts to generate sperm from embryonic stem cells. However, the results are largely preliminary and need further studies to make their conclusions more convincing.

Major comments

1. The evidence for “spermatogenesis” is based only on the expression and relative area of Acr-GFP or PNA. An increase in the GFP-positive area can be attributed to proliferation of germ cells, facilitated spermatogenic progression, or loss of GFP-negative cell populations. Reference #16 was given to

justify the experimental method, but it was published in 1999 from a separate laboratory. More vigorous evaluation of spermatogenesis, such as morphology with high resolution, is required. Generation of fertile spermatids supports the occurrence of complete spermatogenesis, but only two seminiferous tubules were tested. No control was provided to assess whether the success rates of 6–12% were low or high. The conclusion such as “RTI supplementation improves the efficiency of *in vitro* spermatogenesis in fetal testes” (Lines 127-128) is not well supported.

Reply: Thank you for these critical and fundamental comments regarding our quantification methods and the validity of our conclusions. We have carefully addressed each concern as follows.

- **Quantification of GFP-positive area**

We agree that an apparent increase in the relative GFP-positive area could theoretically result from loss of GFP-negative cells rather than true spermatogenic progression. To address this, we performed an additional analysis using the *absolute* GFP-positive area (μm^2) instead of relative percentages (new Supplementary Fig. 2A). This analysis confirmed a significant increase in absolute GFP-positive area in the hypoxia + AEC group, supporting the interpretation that RTI supplementation facilitates spermatogenic progression rather than simply reflecting a relative shift due to cell loss.

- **High-resolution morphological evaluation**

Recognizing the importance of morphology, we have added high-resolution immunostaining images (new Supplementary Fig. 2C) that clearly show numerous elongating spermatids with typical morphology. We believe these images provide the stronger morphological validation you requested.

- **Functional validation and success rate of ROSI**

We conducted microinsemination (ROSI) to functionally validate spermatids produced *in vitro* from E12.5 fetal testes. The success rates cited by the reviewer (6–12%) appear to be calculated using injected oocytes as the denominator (from Table S1). Our standard practice is to report the success rate per transferred 2-cell embryos, which yielded 8.7% (2/23) and 22.4% (13/58) in this study. Although we did not include an *in vivo*-derived round spermatid control in the present experiment, the reported success rates are consistent with the established range for ROSI in mice: ~30% on average, with a reported span from 5% to 47% (Ogonuki et al., 2010 & 2011; DOI: [10.1371/journal.pone.0011062](https://doi.org/10.1371/journal.pone.0011062), DOI: [10.1262/jrd.11-008m](https://doi.org/10.1262/jrd.11-008m)). Our previous studies using *in vitro*-derived spermatids showed comparable or lower rates—2.6% (Yokonishi 2014; DOI: [10.1038/ncomms5320](https://doi.org/10.1038/ncomms5320)), 5% (Ishikura 2021; DOI: [10.1016/j.stem.2021.08.005](https://doi.org/10.1016/j.stem.2021.08.005)), 12.5% and 43.8% (Sato 2011; DOI: [10.1038/nature09850](https://doi.org/10.1038/nature09850)), indicating that the present results fall within the expected and accepted range.

- **Basis for our conclusions**

Our conclusion that RTI supplementation improves the efficiency of *in vitro* spermatogenesis is

supported by the quantitative GFP data (Fig. 4b, 4c) and the new absolute area analysis (Supp. Fig. 2A). Separately, our conclusion that complete spermatogenesis was achieved is supported by both high-resolution morphological evidence of elongating spermatids (Supp. Fig. 2C) and the ultimate functional proof of fertility through ROSI.

2. As the authors pointed out, “The mechanism by which RTIs promote spermatogenesis remains unclear” (Line 211) is a major weakness of this paper. Some potential mechanisms, such as inhibition of LINE1, can be easily tested. The authors can also extend the discussions based on the dosage, specificity, and effective time of RTIs. Sertoli cell differentiation can be monitored by expression of a couple of key molecules such as SOX9 and DMRT1.

Reply: Thank you for these constructive comments and for pointing out the importance of exploring potential mechanisms. We have taken concrete steps to strengthen this aspect.

To test the proposed role of retrotransposon suppression, we performed new experiments focusing on LINE-1 activity. The new data (now presented in Figure 6) show that LINE-1 expression and genomic copy number increase during culture but are significantly suppressed by RTI treatment. This provides direct molecular evidence supporting our hypothesis that RTIs promote spermatogenesis through retrotransposon inhibition.

Regarding Sertoli cell markers, we agree that SOX9 and DMRT1 are well-established indicators of Sertoli cell differentiation. However, both genes are already robustly expressed at E12.5, when Sertoli cell fate is determined. Their expression levels remain relatively stable thereafter, making them less sensitive indicators of subtle changes during culture. We therefore considered them less informative for evaluating the effects of RTIs at later culture stages. Instead, we prioritized direct analysis of retrotransposon activity, which is more closely related to our mechanistic hypothesis.

3. The culture conditions need to be better justified. “Fetal Bovine Serum (FBS), a common culture supplement” (Lines 154-155) is not true. 10% FBS is known to promote the growth of fibroblast but not of epithelial cells. Furthermore, the basic medium already contains serum replacement supplements. Adding FBS can be toxic. Did the authors adjust the basic medium with FBS supplementation? The culture was performed at 34°C. This is not the natural environment for fetuses. Did the authors optimize the temperature for the culture of fetal testes?

Reply: We appreciate the reviewer’s insightful comments regarding our culture conditions.

First, we agree that FBS is not an ideal or “general-purpose” supplement for testicular cultures and

have revised the text to avoid implying otherwise. Our intent was to explore whether undefined growth-promoting factors could transiently enhance tissue expansion during the early culture period. Therefore, we supplemented a base medium lacking KnockOut Serum Replacement (KSR) with two concentrations of FBS (2.5% and 10%). As the reviewer correctly pointed out, FBS supported somatic proliferation but did not improve spermatogenesis and in fact reduced the number of differentiated germ cells; these findings are now emphasized in the Results and Discussion. Regarding temperature, we selected 34 °C because previous studies have clearly shown that this condition improves the efficiency of spermatogenesis *in vitro* (Gohbara et al., 2010; DOI: 10.1095/biolreprod.110.083899; Hirano et al., 2022; DOI: 10.1038/s42003-022-03449-y). This temperature has also been effective in our earlier work with fetal mouse testes (Kojima et al., 2016; DOI: 10.1095/biolreprod.116.140277). For this reason, we did not test other temperatures in the present study. We agree, however, that evaluating whether a higher temperature such as 37 °C during the initial culture period might benefit fetal testis growth is an important question for future optimization.

4. Discussion “The fact that spermatogenesis could be induced from E12.5 testes --- suggests that the minimal testicular architecture and somatic support required for germ cell development are already in place at this stage” (Lines 221-223) is ambiguous. All cells there underwent differentiation during culture. This experimental design revealed that all precursor cells are there to support spermatogenesis independent of other parts of the body such as the pituitary.

Reply: Thank you for this insightful and constructive comment. We agree entirely that our original wording was ambiguous and that your suggested interpretation more accurately and powerfully conveys the significance of our findings. We have revised this sentence in the Discussion section to reflect your excellent suggestion. The revised text now reads:

'The successful induction of complete spermatogenesis from E12.5 testes demonstrates that the testicular primordium at this stage contains all the necessary precursor cells to autonomously construct the required architecture and support the entire process *in vitro*, independent of systemic cues from other organs such as the pituitary gland.'

5. Figure 3F: “Quantitation of PNA-positive cells per seminiferous tubule” does not provide correct measurements. Some seminiferous tubules in a section are round and small while others are long. Quantitation must be against a consistent unit size.

Reply: Thank you for this critical point regarding our quantification method. We agree that normalizing to a consistent unit size is a more rigorous approach than quantifying per tubule cross-

section, which can be variable.

To address this, we have performed a new analysis where we normalized the number of PNA-positive cells to the seminiferous tubule area. We have now included this new data as Supplementary Fig. 1. Importantly, this more rigorous quantification yields the same result and supports our original conclusion. This confirms that the observed increase in PNA-positive cells is a genuine biological effect and not an artifact of the initial quantification method. We are grateful for this suggestion, as it has significantly strengthened the robustness of our data.

Minor comments

- The title needs rephrasing.

Reply: We appreciate the reviewer's suggestion regarding the title. After consideration, we would like to retain the current title, "Fetal mouse testes *in vitro* produced fertile spermatids with reverse transcriptase inhibitors." We believe it accurately and concisely conveys the central and novel finding of our study: the successful induction of functional spermatids capable of producing fertile offspring from fetal mouse testes using reverse transcriptase inhibitors.

In the revised version, we have added new data showing that retrotransposon expression is upregulated in fetal testis organ culture and that supplementation with reverse transcriptase inhibitors effectively suppresses this activation. These findings further substantiate the mechanistic rationale for including "reverse transcriptase inhibitors" in the title. We have also clarified this point in the Results and Discussion. We therefore respectfully propose to keep the original title.

- Line 38: "sperm" should be "spermatozoa."

Reply: Thank you for pointing this out. Upon reconsideration, we have changed "sperm/spermatozoa" to "elongating spermatids" throughout the manuscript, except in the explanation of Fig. 4e. We chose this term because "spermatozoa" specifically refers to cells that have detached from Sertoli cells and are released into the lumen of the seminiferous tubule, which is generally not the case in our culture system.

- Line 54: "also reported in those contexts." --- The meaning is unclear.

Reply: Thank you for your feedback. I have revised the introduction as follows (Line 50-52): Subsequently, this method has also been successfully used to induce complete *in vitro* spermatogenesis from other types of tissues, including adult mouse and cryopreserved testicular tissues.

- Line 59: "decreasing developmental stage" is awkward. "earlier developmental stages"?

Reply: We thank the reviewer for pointing this out. I have corrected it to "earlier." (Line 56)

- Line 271: Explain why “Drinking water was acidified.”

Reply: Drinking water was acidified to a pH of 2.5-3.0 with hydrochloric acid. This is a standard husbandry practice to prevent the growth of opportunistic microbial pathogens, such as *Pseudomonas aeruginosa*, in the water supply, thereby maintaining animal health and reducing non-experimental variability.

- Line 298: “goat anti-chicken IgG” --- Chicken does not have IgG.

Reply: Thank you for pointing that out. I have corrected it to IgY.

- Lines 347-348: Give the name of the target gene, Acr-GFP, for PCR genotyping in the text. In Table S2, specify the strand and orientation targeted by each primer and the combination of primers for amplifying WT and TG bands.

Reply: Thank you for pointing that out. I have corrected the method and TableS2.

Reviewer #1 (Remarks to the Author):

The authors have addressed my questions. Thanks for their hard work.

Reply: We appreciate your constructive comments which helped great deal in elaborating the manuscript. Thank you.

Reviewer #3 (Remarks to the Author):

1. **Lines 69 & 181: “immature E12.5 fetal testes” --- The meaning of “immature” is unclear here. It is better to remove it.**

Reply: We agree with the reviewer’s suggestion. The word “immature” has been removed from lines 69 and 181 to improve clarity.

2. **Lines 451 & 483: “cultured with --- medium” should read better “cultured in --- medium.”**

Reply: We have corrected this throughout the manuscript. All instances of “cultured with --- medium” have been changed to “cultured in --- medium” (including lines 451 and 483) to ensure appropriate terminology.